# Visual Scoring of Sacroiliac Joint/Sacrum Ratios of Single-Photon Emission Computed Tomography/Computed Tomography Images Affords High Sensitivity and Negative Predictive Value in Axial Spondyloarthritis

**DOI:** 10.3390/diagnostics13101725

**Published:** 2023-05-12

**Authors:** Eun-Chong Yoon, Jong-Sun Kim, Chae Hong Lim, Soo Bin Park, Suyeon Park, Kyung-Ann Lee, Hyun-Sook Kim

**Affiliations:** 1Division of Rheumatology, Department of Internal Medicine, Soonchunhyang University Seoul Hospital, Soonchunhyang University College of Medicine, Seoul 04401, Republic of Korea; 2Department of Nuclear Medicine, Soonchunhyang University Seoul Hospital, Soonchunhyang University College of Medicine, Seoul 04401, Republic of Korea; 3Department of Biostatistics, Soonchunhyang University Seoul Hospital, Soonchunhyang University College of Medicine, Seoul 04401, Republic of Korea; 4Department of Applied Statistics, Chung-Ang University, Seoul 156-756, Republic of Korea

**Keywords:** spondyloarthritis, sacroiliac joint/sacrum ratio, bone SPECT/CT, MRI

## Abstract

Spondyloarthritis (SpA) is characterized by inflammatory back pain. Magnetic resonance imaging (MRI) was the earlier gold standard technique for detecting early inflammatory change. We reassessed the diagnostic utility of sacroiliac joint/sacrum (SIS) ratios of single-photon emission computed tomography/computed tomography (SPECT/CT) for identifying sacroiliitis. We aimed to investigate of SPECT/CT in diagnosing SpA using a rheumatologist’s visual scoring of SIS ratios assessment. We conducted a single-center, medical records review study of patients with lower back pain who underwent bone SPECT/CT from August 2016 to April 2020. We employed semiquantitative visual bone scoring methods of SIS ratio. The uptake of each sacroiliac joint was compared to that of the sacrum (0–2). A score of 2 for the sacroiliac joint of either side was considered diagnostic of sacroiliitis. Of the 443 patients assessed, 40 had axial SpA (axSpA), 24 being radiographic axSpA and 16 being nonradiographic axSpA. The sensitivity, specificity, and positive and negative predictive values of SIS ratio of SPECT/CT for axSpA were 87.5%, 56.5%, 16.6%, and 97.8%, respectively. In receiver operating curve analysis, MRI better diagnosed axSpA than did SIS ratio of SPECT/CT. Although the diagnostic utility of SIS ratio of SPECT/CT was inferior to MRI, visual scoring of SPECT/CT affords high sensitivity and negative predictive value in axSpA. When MRI is inappropriate for certain patients, SIS ratio of SPECT/CT is an alternative tool for identifying axSpA in real practice.

## 1. Introduction

Spondyloarthritis (SpA) comprises several related but distinct disorders: ankylosing spondylitis, psoriatic arthritis (PsA), enteropathic arthritis/spondylitis, or reactive arthritis [1]. Recently, there has been a unifying proposal for the classification of SpA into axial SpA (axSpA) and peripheral SpA [2,3]. Many epidemiological studies have been conducted for the prevalence of SpA and its subtypes, and results vary widely ranging from 9 to 30 per 10,000 [4]. AxSpA is characterized by chronic inflammatory back pain, peripheral arthritis, and enthesopathies which can lead to structural damage. The most prodromal symptom is insidious lower back pain (LBP) or buttock pain. AxSpA usually starts in the third decade of life with a male-to-female ratio of 2:1 [5]. Patients with axSpA suffer from pain, functional disability, fatigue, and limitation in activities and social participation. Considering most axSpA starts at a young age, socioeconomic burdens such as education, employment, and economic status affected by axSpA are crucial [6]. Currently, HLA B27 remains the best genetic biomarker for diagnosing AS. C-reactive protein (CRP) measures disease activity, predicting structural progression and therapeutic response [7]. Radiographic axSpA (r-axSpA) is usually termed ankylosing spondylitis (AS) [8]. The 1984 modified New York criteria for AS required detection of advanced sacroiliitis in plain radiographs together with any one of three clinical criteria: inflammatory back pain, limited lumbar spine motion, and/or restricted chest expansion [9]. Although these criteria can be easily assessed, they are not appropriate for early diagnosis and responsible management of AS patients. When structural abnormalities become apparent on X-ray, patients with AS typically already have symptoms, such as pain, for several years [5]. In addition, detecting radiographic sacroiliitis is complicated because of inter-reader variability, high measurement error, and low signal–noise ratio [10]. It is well known that detecting radiographic sacroiliitis (sacroiliac joint inflammation) in conventional X-ray is crucial but complicated, leading to the delay of diagnosis in AS. However, as magnetic resonance imaging (MRI) has been used as an additional imaging arm for diagnosing axSpA by 2009 ASAS criteria, sacroiliac joint inflammation can be used as an early diagnostic marker [2]. In several studies using bone biopsies, animal models, and imaging, the earliest detectable change in biopsy specimens is subchondral bone marrow edema visible on MRI [5]. The criteria classify patients lacking radiographic changes on X-ray as suffering from nonradiographic axSpA (nr-axSpA) and use MRI to identify this subtype [3,11]. In this criteria, active sacroiliitis in SpA is defined by the presence of either one bone marrow edema lesion on two consecutive MRI slices or ≥2 lesions on a single slice [2,5]. However, economic reasons, inherent technical limitations, and contraindications may render MRI inappropriate for routine axSpA diagnosis. The need for alternative or additional imaging modalities to diagnose axSpA has been raised.

Axial inflammation, bone destruction, and new bone formation are critical events in the pathophysiology of axSpA [5]. A tracer such as Tc-99m, which shows a high affinity to metabolically active sites, can provide additional information in axSpA. The incorporation of hybrid single-photon emission computed tomography (SPECT)/computed tomography (CT) is potentially more accurate than planar scintigraphic imaging, as it is based on anatomical information evident in the CT scan [12]. The main advantages of SPECT/CT are better attenuation correction, increased specificity, accurate localization of disease, and evaluation of the involvement of adjacent tissues [13]. SPECT/CT allows three-dimensional localization of tracer activity to the sacroiliac joints by volumetric analysis of the joint using volumes of interest [14]. Although the advantage of providing additional anatomical information is clear, the increase in radiation exposure due to additional CT compared to SPECT alone must be considered [15]. In a study, on average, patient exposure from clinical SPECT/CT was estimated to be 7mSv [16]. Limited studies with few patients have shown that SPECT/CT bone imaging may usefully identify sacroiliitis in axSpA patients, and may thus serve as an alternative to MRI [14]. Here, to evaluate a relatively simple method that rheumatologists can use in real-world practice, we investigated the utility of sacroiliac joint/sacrum (SIS) ratios of SPECT/CT in diagnosing axSpA in patients with LBP compared to MRI and conventional sacroiliac X-ray examination at a single institution, using a visual scoring assessment by one rheumatologist.

## 2. Methods

### 2.1. Patient Recruitment

We initially investigated all out-clinic patients who underwent SPECT/CT for the presence of LBP from August 2016 to April 2020 at a single institution. These patients were not only from the rheumatology department, but from all departments, including orthopedic surgery and neurosurgery department. We checked age, sex, body mass index (BMI), pain duration (months), smoking history, clinical symptoms (numbness, claudication, peripheral arthritis, Achilles tendinitis, enthesitis, uveitis, and psoriasis), the erythrocyte sedimentation rate (ESR), the C-reactive protein (CRP) level, and HLA B27 status at initial records. ESR, CRP, and HLA B27 was conducted at the same institution. Bath Ankylosing Spondylitis Disease Activity Index (BASDAI) score was analyzed in only axSpA. Activity in SpA refers to the inflammation caused by the disease, commonly assessed in daily practice with the BASDAI [1,5]. The BASDAI contains six questions addressing fatigue, back pain, peripheral joint pain/swelling, enthesitis, and morning stiffness. The study was conducted in accordance with the Declaration of Helsinki and was approved by the Institutional Review Board for Human Research, Soonchunhyang University Seoul Hospital (approval no. 2020-05-008). The requirement to obtain informed consent was waived; this medical records review study involved a minimum risk to enrolled patients and all patients were anonymized.

### 2.2. Clinical Assessment

Patients who met the 2009 ASAS criteria for axSpA formed the disease group and others were in the control group. Nr-axSpA is a subtype of axSpA, and its classification was based on the MRI or clinical characteristics. The final diagnosis was confirmed by a single trained rheumatologist who reviewed the medical charts, laboratory test results, and SIS ratios of SPECT/CT. The diagnosis of active sacroiliitis on MRI was adopted based on the new ASAS classification criteria [11]. The final reading of hip joint MRI and conventional sacroiliac X-ray confirmed by radiologists was reviewed by a single rheumatologist to diagnose ax-SpA.

### 2.3. Acquisition of SPECT/CT and Visual Scoring of SIS Ratios

All imaging data were acquired using a Symbia Intevo16 hybrid SPECT/CT system (Siemens Healthineers, Erlangen, Germany) featuring an integrated, dual-head SPECT camera with a 16-slice helical CT scanner [17,18]. In all patients, the field of view (FOV) completely covered the pelvic region. Tc-99m methylene diphosphonate (MDP) was injected (740 MBq, 20 mCi) and SPECT was performed 3 h later. The phosphonate groups on Tc-99m MDP bind firmly to hydroxyapatite crystals in bone through chemisorption [19]. Because of its high affinity to metabolically active sites in bone, Tc-99m MDP is commonly used in scintigraphic uptake studies to identify the high active lesion [20]. The imaging parameters were low-energy high-resolution collimator, matrix 256 × 256, step-and-shoot mode, and the use of dual-head gamma cameras with 45 steps/detector (angle 4°) at 22 s/step. After SPECT acquisition, CT images were generated at 110 kV and 80 mA using the Auto-mA mode with a 256 × 256 matrix and a pitch of 1.2, and reconstructed into 1.5 mm thick slices [21]. Finally, the SPECT and CT images were coregistered to yield fusion SPECT/CT images.

Regarding SIS ratios of SPECT/CT visual scoring, a score of 0 was assigned when tracer uptake by the sacroiliac joint was less than that of the sacrum, 1 when the uptakes were equal, and 2 when the sacroiliac joint uptake was greater than that of the sacrum (Figure 1). Both sacroiliac joints were scored. A visual score of 2 on either side was taken to indicate sacroiliitis in SPECT/CT. A single rheumatologist performed all scoring of SIS ratios of SPECT/CT; this professional had been trained and checked by an experienced nuclear physician [22]. Even if one side scored only 2 points of SIS ratios of SPECT/CT visual scoring, the diagnostic criteria were satisfied, but for the classification of the strongly positive subgroup, it was defined as a group with 2 points on both sides. Additional information other than SIS ratios from SPECT/CT was not used in these investigation.

### 2.4. Statistical Analyses

Statistical analysis employed SPSS software for Windows (version 26.0; SPSS, Inc. Chicago, IL, USA) and Rex (version 3.6.0, RexSoft Inc., Seoul, Korea). We compared the demographic and clinical characteristics and laboratory findings of axSPA and control subjects. Continuous parameters (age, the BMI, pain duration, the ESR, and the CRP level) were compared using the Mann–Whitney *U*-test. Data are expressed as medians (Q1, Q3) for continuous variables and as absolute frequencies with percentages (%) for qualitative variables. Categorical parameters including sex, clinical symptoms (numbness, claudication, peripheral arthritis, Achilles tendinitis, enthesitis, uveitis, and psoriasis), and HLA-B27 status were compared using the chi-square or Fisher’s exact test. The diagnostic performances of SIS ratios of SPECT/CT, MRI, and X-ray imaging were evaluated by calculating the sensitivity, specificity, positive predictive value (PPV), and negative predictive value (NPV) afforded by receiver operating characteristic (ROC) analyses. The areas under the curves (AUCs) with 95% confidence intervals were measured for all quantitative indices of SIS ratios of SPECT/CT, MRI, and conventional X-ray imaging. During AUC comparisons, all missing values (all methods) were excluded. Statistical significance (all analyses) was set to *p* < 0.05.

## 3. Results

### 3.1. Patient Characteristics

A total of 443 patients were included. Among the 443 cases that took SPECT/CT scans with LBP, axSpA was diagnosed in 40 patients: their baseline demographic and clinical characteristics are summarized in Table 1. All 443 patients underwent SPECT/CT, 372 underwent sacroiliac viewing X-ray, and only 161 underwent MRI. The study involved 182 men and 261 women. Of 40 patients with axSpA, 24 (60%) had r-axSpA while 16 had nr-axSPA. The mean age of the 40 axSpA patients was 38.5 years and that of controls was 56 years (*p* < 0.001). No significant between-group differences were evident in terms of the smoking history. Twelve patients in the axSpA group and 149 patients in the control group performed MRI. A total of 29 patients in the axSpA group and 72 patients in the control group performed HLA-B27 test. The number of patients who performed both MRI and HLA-B27 was 8 in the axSpA group and 33 in the control group. A total of 208 patients in the control group did not perform both MRI and HLA-B27. Of 443 patients, 71 did not perform sacroiliac X-rays, but these were all in the control group.

### 3.2. Clinical Features of axSpA According to SIS Ratios of SPECT/CT Visual Scoring

An SIS ratio of SPECT/CT score of 2 for either sacroiliac joint was taken to indicate sacroiliitis. As the proportion of positive patients was too large using this criterion, we compared the clinical features of a strongly positive subgroup and the other group. Thirty-two axSpA patients had summed scores of 4 (strongly positive, positive on both sides), whereas eight had summed scores less than 4. There were no significant differences in any demographic feature, clinical symptom, or laboratory finding between the strongly positive subgroup and the nonpositive group. Of patients with visual scores of 4, all who underwent MRI evidenced bone marrow edema, whereas only 50% of patients with visual scores less than 4 exhibited edema. However, statistical significance was lacking (*p* = 0.109, Table 2).

### 3.3. Use of Sacroiliac Images for Diagnosis of axSpA and Nonradiographic axSpA

The sensitivity and specificity of SIS ratios of SPECT/CT for axSpA were 87.5% and 56.5% and the PPV and NPV were 16.6% and 97.8%, respectively, consistent with nr-axSpA status. The figures for the nr-axSpA patients were 86.6%, 55.2%, 14.8%, and 97.8%, respectively. SIS ratios of SPECT/CT were more sensitive than MRI in terms of diagnosing axSpA (87.5%) and nr-axSpA (86.6%); the NPV (97.8%) was comparable to that of MRI. The diagnostic utilities of SIS ratios of SPECT/CT, MRI, and X-ray imaging in axSpA and nr-axSpA patients are compared in Table 3. In ROC curve analysis, the order was SIS ratios of SPECT/CT (AUC = 0.665; 95% CI 0.539–0.798), X-ray imaging (AUC = 0.829; 95% CI 0.82–90.927), and MRI (AUC = 0.898; 95% CI 0.765–0.991); only SIS ratios of SPECT/CT and MRI evidenced a statistically significant difference (*p* = 0.043, Figure 2).

## 4. Discussion

The main objective of our study was to analyze the role of visual scoring of SIS ratios of SPECT/CT in the diagnosis of axSpA by a rheumatologist. Sacroiliac joint inflammation is an important criterion when diagnosing axSpA. As most cases are young, early diagnosis is critical; appropriate treatment preserves productivity. “Treat-to-target (T2T)” is a major principle in treating many diseases, such as inflammatory rheumatic diseases, including rheumatoid arthritis and SpA. The objectives of T2T are to ensure accurate and reliable diagnosis with more reasonable treatment decisions. Plain radiography has been commonly used as a diagnostic tool, and was considered essential in the 1984 modified New York criteria. However, it detects sacroiliitis only a few years after symptom onset [23]. The ASAS SpA criteria consider MRI the gold standard, allowing accurate detection of early cartilaginous and bony changes [24]. However, MRI is contraindicated in patients with pacemakers, cochlear or metal implants, and claustrophobia; it is also expensive [14]. Such patients may require another imaging technique for diagnosis of early sacroiliitis in clinical settings.

SPECT/CT of bone has been used to assess various musculoskeletal diseases; the technique yields functional and locational information, and reveals specific structural patterns. It has also been employed to assess injuries caused by repetitive exercise or activity (particularly in athletes) [25] and hand, wrist, and ankle pain [12]. Lim et al. studied 84 patients and found that the hand and wrist standardized uptake values (SUVs) in SPECT/CT images differed reliably from those with different diseases [26]. Kim et al. used SPECT/CT to evaluate knees; the values were highly correlated with the traditional imaging parameters used to explore medial compartment osteoarthritis severity [27].

As SPECT/CT serves as a complementary imaging modality for patients with various diseases, there is an increasing need for quantitative or semiquantitative SPECT/CT. When this was used to explore osseous metabolism, data derived via simple visual and quantitative analyses (SUVs for the latter) differed [27]. Efforts to use SPECT/CT in a quantitative manner to evaluate back pain continue. Ornilla et al. found that quantitative SPECT/CT alone afforded moderate sensitivity (51%) and high specificity (86%) in a study of patients with LBP; the addition of MRI improved both sensitivity (67%) and specificity (86%), being superior to those of MRI alone [14]. Parghane et al. found good agreement and correlation between SPECT/CT with reference standard and MRI in detection of sacroiliitis [24]. Kim et al. used SPECT/CT to derive SIS ratios, and successfully detected subtle changes in early axSpA with sensitivity of 80.0% and specificity of 84.6% (in contrast to bone scintigraphy) [21]. Another study found that semiquantitative analysis of SPECT/CT using SIS ratio can be an useful predictor in identifying early axSpA patients who are at high risk of structural progression [28]. In assessing activity of sacroiliitis with SPECT/CT, Lee et al. found that SIS ratio showed more favorable results than SUV parameters such as SUVmax [29].

We also used a visual scoring method based on the SIS ratio; this is appropriate in real-world clinical practice by rheumatologists. In the control group, 230 patients showed negative results in SIS ratios of SPECT/CT scan, and 92 patients among them underwent MRI, while all those patients showed negative results. On the other hand, in both the axSpA and control group, no patients showed positive result from SIS ratios of SPECT/CT scans and negative MRI results simultaneously. Our principal objective was to explore the diagnostic utility of SIS ratios of SPECT/CT visual scoring in terms of axSpA diagnosis among patients with LBP. Unlike previous studies, in our research, PPV, AUC, specificity, and accuracy of SIS ratios of SPECT/CT visual scoring judged by a rheumatologist were lower than those of MRI. In the axSpA group, five patients showed negative results in SPECT/CT scan. Among them, three patients underwent MRI, and one showed positive MRI results. In such patients, SIS ratios of SPECT/CT afforded a higher sensitivity and NPV than MRI; performance was good. Even in the 16 nr-axSpA patients, it performed well and consistently. In previous studies, 10–40% of patients with nr-axSpA developed structural changes in the sacroiliac joints within the first 2–10 years of disease onset. In the same study, the estimated chance of progression from nr-axSpA to axSpA was over 50% [30]. Nr-axSpA has been considered an early stage of axSpA, but nr-axSpA itself has clinical importance. Disease activity levels and functional impact of the Nr-axSpA patients are similar to AS patients; the disease burdens appear equal [2]. It is already known from other studies that nr-axSpA patients show diagnostic delay compared to axSpA patients [31].

While the importance of diagnosing nr-axSpA has been raised for earlier treatment, the diagnosis of nr-axSpA is still challenging. Inflammatory changes on MRI can also appear in mechanical problems, and the need for additional information has been raised [32]. Given the consistent diagnostic performance of SIS ratios of SPECT/CT in terms of nr-axSpA, the technique can serve as a complementary modality for early diagnosis and management of nr-axSpA patients.

The present study had certain limitations. First, although we initially included 443 patients with LBP, only 40 were diagnosed with axSpA. Our study used a rather modest number of retrospective medical records analysis from single center, so it was impossible to obtain the number of samples with statistical significance. The significantly small number of analyses in the SpA group is a limitation of this study. The final number of SpA patients for analysis yield needs to be increased to make a firm conclusion. Second, some patients with LBP had been diagnosed with axSpA prior to SPECT/CT. It was also highly selected by clinicians, which might have influenced the results as selection bias. Third, a single expert rheumatologist assessed all the images without any valid data provision. Above all, the most unfortunate thing about this study is that the important strength of SPECT/CT was not utilized. The advantage of SPECT/CT over bone scintigraphy is that SPECT/CT shows a fusion image of nuclear image and CT, providing both functional and locational information. In some cases, CT images can provide additional information to distinguish chronic changes such as erosion, joint space alterations, and subchondral sclerosis in the joint [14,24]. Conjoining anatomical information from CT images can yield better characterization of equivocal lesion on SPECT [33,34]. In addition, measuring the area of the affected SI joint on the SPECT/CT can lead to better estimation of SIS ratio [21]. However, the present study only used a tracer uptake of SIS ratio by a visual assessment, which could not fully offer the advantage of SPECT/CT.

## 5. Conclusions

When MRI is inappropriate or unavailable in the real world, visual scoring of SIS ratios of SPECT/CT may serve to identify axSpA with high sensitivity and a high NPV. For those patients with an increased likelihood, based on clinical considerations such as peripheral arthritis, uveitis, etc., SIS ratios of SPECT/CT scanning may also provide support for the diagnosis of nr-axSpA in the early disease phase.

## Figures and Tables

**Figure 1 diagnostics-13-01725-f001:**
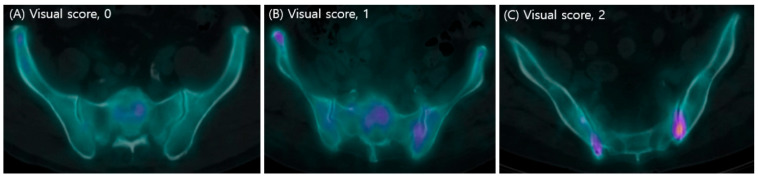
The representation of SIS ratios of SPECT/CT visual scoring: (**A**) a score of 0 was assigned when tracer uptake by the sacroiliac joint was less than that of the sacrum, (**B**) 1 when the uptakes were equal, and (**C**) 2 when the sacroiliac joint uptake was greater than that of the sacrum.

**Figure 2 diagnostics-13-01725-f002:**
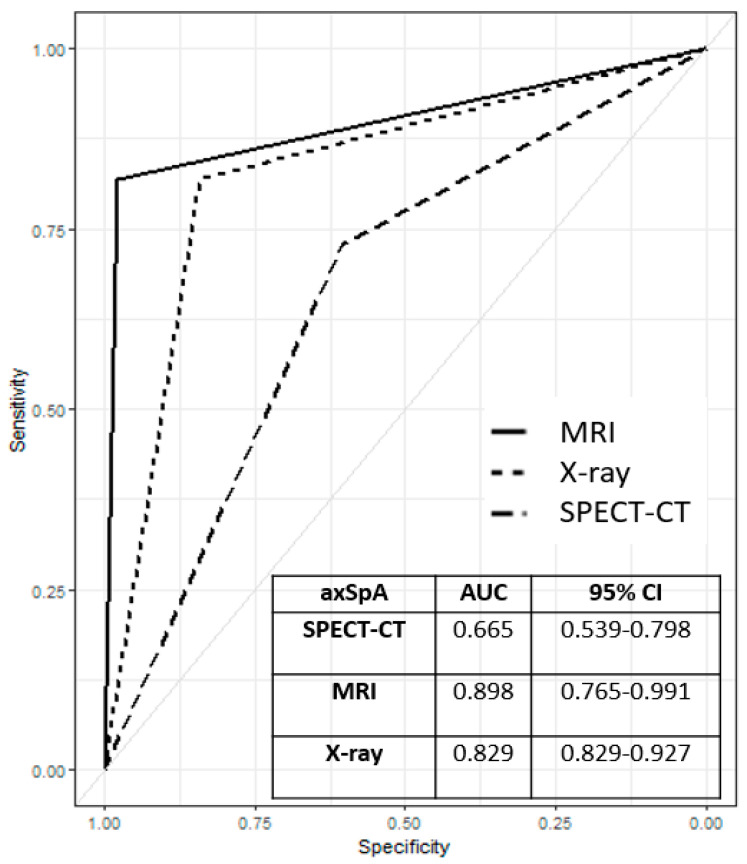
The receiver operating characteristic curves of SIS ratios of SPECT/CT, MRI, and X–ray imaging used to diagnose axSpA. All missing values (all methods) were excluded. AxSpA: axial spondyloarthritis, AUC: area under the receiver operating characteristic curve, SPECT–CT: SIS ratios of single-photon emission computed tomography/computed tomography, MRI: magnetic resonance imaging.

**Table 1 diagnostics-13-01725-t001:** Baseline characteristics using medical records review.

	AxSpA (*n* = 40)	Control (*n* = 403)	*p*-Value
Diagnosis, *n* (%)			
Radiographic axSpA	24 (60)		
Nonradiographic axSpA	16 (40)		
Demographics			
Age	38.5 (22, 51.25)	56 (44, 64.5)	<0.001 *
Sex, *n* (%)			
Male	24 (60)	158 (39.20)	0.011 *
Female	16 (40)	245(60.79)	
BMI (kg/m^2^)	22.15 (20.02, 23.82)	23.8 (21.4, 26.48)	0.018 *
Pain duration (months)	24 (8, 48)	6 (1, 17)	<0.001 *
Ever smoker, *n* (%)	5 (12.5)	48 (11.91)	0.804
Clinical symptoms, *n* (%)			
Numbness ‡	11 (27.5)	124 (30.76)	0.661
Claudication ‡	5 (12.5)	53 (13.1)	0.903
Peripheral arthritis ‡	4 (10)	2 (0.49)	0.001 *
Achilles tendinitis ‡	5 (12.5)	2 (0.49)	<0.001 *
Enthesitis ‡	5 (12.5)	1 (0.24)	<0.001 *
Uveitis ‡	5 (12.5)	5 (1.24)	<0.001 *
Psoriasis ‡	1 (2.5)	2 (0.49)	0.249
Laboratory finding			
HLA B27 ‡, *n* (%)	21/29 (72.41)	30/72 (41.67)	0.010 *
ESR (mm/hr)	20 (12, 47)	28 (16, 47)	0.072
CRP (mg/dL)	0.07 (0.03, 0.34)	0.12 (0.04, 0.61)	0.274

*n*: Numbers, AxSpA: axial spondyloarthritis, BMI: body mass index, ESR: erythrocyte sedimentation rate, CRP: C-reactive protein. Data are expressed as medians (Q1, Q3) for continuous variables, and as absolute frequencies and percentages (%) for qualitative variables. Continuous parameters were compared using the Mann–Whitney *U*-test. ‡: Analyzed employing the Fisher exact test. * *p* < 0.05.

**Table 2 diagnostics-13-01725-t002:** Comparison of clinical features of axSPA among patients strongly positive and others in SIS ratios of SPECT/CT visual scoring.

	Sum Score = 4 (*n* = 32)	Sum Score < 4 (*n* = 8)	*p*-Value
Demographics, median (Q1, Q3)			
Age	34 (21.5, 50.25)	44 (36.0, 53.25)	0.187
Sex (*n* of male, %)	18 (56.25)	6 (75)	0.439
BMI (kg/m^2^)	22.15 (19.67, 23.4)	23.85 (20.65, 29.82)	0.457
Pain duration (months)	24 (8.75, 48)	36 (15, 48)	0.726
Ever smoker, *n* (%)	3 (9.37)	2 (25)	0.257
Radiographic axSpA, *n* (%)	18 (56.25)	6 (75)	0.432
Clinical symptoms, *n* (%)			
Numbness	9 (28.12)	2 (25)	1.000
Claudication	3 (9.37)	2 (25)	0.232
Peripheral arthritis	4 (12.5)	0 (0)	0.566
Achilles tendinitis	5 (15.62)	0 (0)	0.563
Plantar fasciitis	4 (12.5)	1 (12.5)	1.000
Uveitis	4 (12.5)	1 (12.5)	1.000
Psoriasis	1 (3.12)	0 (0)	1.000
Laboratory finding, median (Q1, Q3)			
ESR (mm/h)	19 (8, 47)	22.5 (15.25, 59.5)	0.416
CRP (mg/dL)	0.08 (0.03, 0.79)	0.06 (0.025, 0.265)	0.637
BASDAI (0–10), median (Q1, Q3)	4.9 (3.32, 6.77)	5.1 (2.77, 0.67)	0.963
Bone marrow edema in MRI, *n* (%)	7/7 (100)	2/4 (50)	0.109

*n*: Numbers, AxSpA: axial spondyloarthritis, BMI: body mass index, ESR: erythrocyte sedimentation rate, CRP: C-reactive protein level, BASDAI: Bath Ankylosing Spondylitis Disease Activity Index, MRI: magnetic resonance imaging. A score of 2 for either sacroiliac joint was considered to reflect sacroiliitis. In the strongly positive group, both sides were positive (summed score 4). Continuous parameters were compared using the Mann–Whitney *U*-test.

**Table 3 diagnostics-13-01725-t003:** The utility of SIS ratios of SPECT/CT, MRI, and sacroiliac X-ray images for diagnosis of axSpA and nonradiographic-axSpA (nr-axSpA).

axSpA	*n*	AUC (95% CI)	Sensitivity	Specificity	PPV	NPV	Accuracy
SPECT/CT	443	0.720 (0.669–0.774)	0.875 (0.732–0.958)	0.565 (0.515–0.614)	0.166 (0.118–0.224)	0.978 (0.950–0.993)	0.593 (0.546–0.639)
MRI	161	0.899 (0.772–0.991)	0.818 (0.482–0.977)	0.98 (0.942–0.995)	0.75 (0.428–0.942)	0.986 (0.952–0.998)	0.968 (0.929–0.989)
X-ray	372	0.732 (0.654–0.808)	0.589 (0.421–0.774)	0.873 (0.833–0.907)	0.353 (0.239–0.482)	0.947 (0.916–0.969)	0.844 (0.803–0.879)
nr-axSpA							
SPECT/CT	363	0.709 (0.644–0.780)	0.866 (0.692–0.962)	0.552 (0.497–0.606)	0.148 (0.099–0.210)	0.978 (0.946–0.996)	0.578 (0.525–0.629)
MRI	153	0.878 (0.755–0.993)	0.777 (0.399–0.971)	0.979 (0.940–0.995)	0.7 (0.347–0.933)	0.986 (0.950–0.998)	0.967 (0.925–0.989)

AxSpA: axial spondyloarthritis, *n*: number, PPV: positive predictive value, NPV: negative predictive value, AUC: area under the receiver operating characteristic curve, SPECT-CT: SIS ratios of single-photon emission computed tomography/computed tomography, MRI: magnetic resonance imaging.

## Data Availability

Data available upon reasonable request.

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
