# Peer review of "Visual Scoring of Sacroiliac Joint/Sacrum Ratios of Single-Photon Emission Computed Tomography/Computed Tomography Images Affords High Sensitivity and Negative Predictive Value in Axial Spondyloarthritis"

_diagnostics, 2023, doi:10.3390/diagnostics13101725_

Round 1

Reviewer 1 Report

MDPI

Diagnostic

Title: Visual scoring of sacroiliac joint/sacrum ratios of single-photon emission computed tomography/computed tomography images affords high sensitivity and negative predictive value in axial spondyloarthritis.

Abstract: Objective of the study must be made more clearer

Introduction:

1)    Information of spondyloarthritis and its difference from  axial spondyloarthritis, prevalence of the disease, whether females are more affected than male etc information must be incorporated along with references.

2)    Objective of this study must be well elaborated. Out of various forms of spondyloarthritis, why  only axial spondyloarthritis was selected to be studied?

3)    How sacroiliac joints are most important for axSpA. Reference for the line  “The sacroiliac joints are ….. currently used” must be provided.

4)    What are the available marker/biomarker to detect/diagnose the various forms of spondyloarthritis. Is sacroiliac joint inflammation is used as early diagnostic marker of axSpA. Reference must be provided.

5)    Importance, adavnatge/disadvantage of SPECT/CT  must be incorporated along with references.

6)    How volumetric analysis of the joint helps in pain management. Is it specific to axSpaA?

7)    SPECT/CT bone imaging  identifies sacroiliitis, sacroiliac joint/sacrum (SIS) ratio in axSpA patients. However its usefulness for diagnosing asSpA must be elaborately incorporated along with references.

Method:

1)    How BASDAI score helps in diagnosing asSpA. The importance of BASDAI must be incorporated along with references.

2)    Why Nr-axSpA was considered in this study?

3)    What are the criteria to diagnose active and inactive sacroiliitis?

4)    How new ASAS classification was helpful in diagnosis of axSpA disease?

5)    Reference must be provided for Symbia Intevo16 hybrid SPECT/CT system

6)    What is the role of MDP. Why and how much concentration was injected to the asSpA. Incorporate information with references

7)    It is mentioned that the additional information other than SIS ratios from SPECT/CT were  not investigated in the said study. What are those factors and why those factors were not considered. Please justify.

8)    What was the statistics used to decide the number of control and diseased subjects. Which ratio gives the best results. In the said study diseased samples were 40 whereas control samples were 403 (table1). Please justify the number of samples (disease and control) considered along with references.

Results:

1)    Sacroiliac joint and sacrum must be labelled in Fig1 in order to have clear understanding of the score  (0,1,or 2). How sacroiliitis was identified in SPECT/CT appropriately.

2)    It is written “A total of 443 patients were included”. However table 1 shows AxSpA (n=40) indicating diseased case is 40 and Control subjects were n=403.  What were the criteria considered for control sample.  Further, total number of cases in SPECT/CT, MRI, and sacroiliac X-ray images  is different.

Author must clarify about the total number of cases and control.

3)    The data (table1) shows all male subjects. Why females were not included in the study. Female participants must have been involved. Please include RF, duration of disease, medication etc in the table.

4)    The disease duration is not mentioned. Therefore the data ie “duration of pain in months” is not clear. Pain must have been mentioned in number of days per year.Please clarify.  

5)    Comparison of data between SPECT/CT and MRI  must be  have been shown in table format for more clear analysis.

6)    Figure showing MRI evidenced bone marrow edema could have been incorporated.

Discussion:

1)    In   axSpA,  apart from  Sacroiliac joints there are other responsible factors causing axSpA and SPECT/CT bone imaging identifies sacroiliitis in axSpA patients. Sufficient discussion is needed in this regard  along with references.

2)    Utility of sacroiliac joint/sacrum (SIS) ratios for the disease management of axSpA must be discussed elaborately along with references.

3)    What are SUV parameters and how does SUV parameters showed more favourable results than SIS ratio. Please discuss.

4)    Correlation between the lab clinical data (ESR, CRP), clinical symptoms (numbness, Claudication etc) and  SIS ratios of SPECT/CT visual scoring could have been generated for the enhancement of axSpA diagnosis.

5)    The advantage/disadvantage of SPECT/CT visual scoring method must have been discussed elaborately  along with references.

Conclusion:

The study objective was diagnostic performance of SIS ratios of SPECT/CT in axSpA. But, very less number of patients  were considered. Comparative analysis was not clear, not validated, and hence conclusion is not sufficiently strong.

Author Response

First of all, thank you for the reviewers’ thoughtful comments. This is the point-by-point answer for the words.

Reviewer1

First of all, thank you for the reviewers’ thoughtful comments. This is the point-by-point answer for the words.

Abstract: Objective of the study must be made more clearer

  • Thank you for the thoughtful review. To clarify the objective of the study,
  • “ In the clinical setting, some patients are impossible to undergo MRI for routine axSpA diagnosis. As need for alternative imaging modality for diagnosis has been raised” is added to the abstract (line 49-51).

Introduction:

  • Information of spondyloarthritis and its difference from axial spondyloarthritis, prevalence of the disease, whether females are more affected than male etc information must be incorporated along with references.
  • The introduction adds information about SpA, its epidemiology, and additional details about axSpA.
  • “ Spondyloarthritis (SpA) comprises several related but distinct disorders: ankylosing spondylitis, psoriatic arthritis (PsA), enteropathic arthritis/spondylitis, or reactive arthritis (1). Recently, there has been a unifying proposal for the classification of SpA into axial SpA (axSpA) and peripheral SpA (2, 3). Many epidemiological studies have been conducted for the prevalence of SpA and its subtypes, results vary widely ranging from 9 to 30 per 10,000 (4).”(in line 68-73)
  • “AxSpA usually starts in the third decade of life with a male-to-female ratio of 2:1 (5).” (in line 75-76)
  • Objective of this study must be well elaborated. Out of various forms of spondyloarthritis, why only axial spondyloarthritis was selected to be studied?
  • Thank you very much for this comment. To clarify the objective of this study, various forms of spondyloarthritis is firstly described, epidemiology of axSpA and its impact on socio-economic status is described.
  • “AxSpA usually starts in the third decade of life with a male to female ratio of 2:1.(5) Patients with axSpA suffer from pain, functional disability, fatigue, and limitation in activities and social participation. Considering most axSpA starts at a young age, socio-economic burdens such as education, employment, and economic status affected by axSpA are crucial (6).”(in line 75-79)
  • How sacroiliac joints are most important for axSpA. Reference for the line “The sacroiliac joints are ….. currently used” must be provided.
  • Thank you very much for this comment.
  • 1984 modified New York criteria for AS requires detection of advanced sacroiliitis, and the imaging arm of 2009 ASAS criteria also focuses on sacroiliac joints. Also, the most prodromal symptom of axSpA is LBP and buttock pain. Reading your thoughtful comment, the meaning of sentence “The sacroiliac joints are … currently used” seems unclear. The sentence is deleted in the revised version. (in line 81-83)
  • What are the available marker/biomarker to detect/diagnose the various forms of spondyloarthritis. Is sacroiliac joint inflammation is used as early diagnostic marker of axSpA. Reference must be provided.
  • Thank you very much for this comment. Information about important marker and biomarkers are described in revised version.
  • “Currently, HLA B27 remains the best genetic biomarker for diagnosing AS. C-reactive protein (CRP) measures disease activity, predicting structural progression and therapeutic response.”(In line 79-81)
  • It is well known that detecting radiographic sacroiliitis (sacroiliac joint inflammation) in conventional X-ray is crucial but complicated, leading to the delay of diagnosis in AS. But as MRI has been used as an additional imaging arm for diagnosing axSpA by 2009 ASAS criteria, sacroiliac joint inflammation can be used as an early diagnostic marker. In some studies with animal models, bone marrow edema of MRI was the earliest detectable change.(in line 91-95)
  • In several studies using bone biopsies, animal models, and imaging, the earliest detectable change in biopsy specimens is subchondral bone marrow edema visible on MRI. Also, detecting radiographic sacroiliitis is complicated because of inter-reader variability, high measurement error, and low signal-noise ratio. Sacroiliac joint inflammation can be detected via magnetic resonance imaging (MRI) in patients with symptoms of AS even when the joints do not appear abnormal in conventional radiography. The criteria classify patients lacking radiographic changes on X-ray as suffering from non-radiographic axSpA (nr-axSpA) and use MRI to identify this subtype.
  • Importance, adavnatge/disadvantage of SPECT/CT must be incorporated along with references.
  • Thank you very much for this comment.
  • The most important advantage of SPECT/CT would be localization of tracer activity and disadvantage would the radiation exposure. In introduction, specific description of advantage and disadvantage of references is added.(in line 111-113)
  • “ A tracer such as Tc-99m, which shows a high affinity to metabolically active sites, can provide additional information in axSpA. The incorporation of hybrid single-photon emission computed tomography (SPECT)/computed tomography (CT) is potentially more accurate than planar scintigraphic imaging, as it is based on anatomical information evident in the CT scan (12). The main advantages of SPECT/CT are better attenuation correction, increased specificity, accurate localization of disease, and evaluation of the involvement of adjacent tissues.(13) SPECT/CT allows three-dimensional localization of tracer activity to the sacroiliac joints by volumetric analysis of the joint using volumes of interest (14). Although the advantage of providing additional anatomical information is clear, the increase in radiation exposure due to additional CT compared to SPECT alone must be considered.(15).” (In line 108-117)
  • How volumetric analysis of the joint helps in pain management. Is it specific to axSpaA?
  • Thank you very much for this comment.
  • The quantification trial of SPECT/CT has been continuously tried in various aspects. Rather than direct pain management, it is a variety of imaging trials to evaluate inflammation and activity.
  • SPECT/CT bone imaging identifies sacroiliitis, sacroiliac joint/sacrum (SIS) ratio in axSpA patients. However its usefulness for diagnosing asSpA must be elaborately incorporated along with references.

-     Thank you very much for this comment

  • Limited studies with few patients have shown that SPECT/CT bone imaging may usefully identify sacroiliitis in axSpA patients, and may thus serve as an alternative to MRI (14). Here, to evaluate a relative simple method that rheumatologists can use in real-world practice, we investigated the utility of sacroiliac joint/sacrum (SIS) ratios of SPECT/CT in diagnosing axSpA in patients with LBP compared to MRI and conventional sacroiliac X-ray examination. (line 118-123)

Method:

  • How BASDAI score helps in diagnosing asSpA. The importance of BASDAI must be incorporated along with references.

- Thanks for your comment.

- The purpose of collecting BASDAI from axSpA patients is to evaluate the activity of the disease after diagnosis. This is clarified in the sentence below.

-“Bath Ankylosing Spondylitis Disease Activity Index (BASDAI) score was analyzed in only axSpA. Activity in SpA refers to the inflammation caused by the disease, commonly assessed in daily practice with the BASDAI (1, 5). The BASDAI includes six questions addressing fatigue, back pain, and so on. became. , peripheral joint pain/swelling, enthesitis and morning stiffness.”(line 135-139)

2)    Why Nr-axSpA was considered in this study?

- Thanks for your comment.

- In previous studies, nr-axSpA has been considered as early stage of axSpA. But recent studies suggest that nr-axSpA itself has clinical importance as equivocal to axSpA. This is added to discussion.

- “In previous studies, 10-40 % of patients with nr-axSpA develop structural changes in the sacroiliac joints within the first 2-10 years of disease onset. In the same study, the estimated chance of progression from nr-axSpA to axSpA was over 50 % (30). Nr-axSpA has been considered an early stage of axSpA, but nr-axSpA itself has clinical importance. Disease activity levels and functional impact of the  Nr-axSpA patients are similar to AS patients; the disease burdens look equal (2).”(in line 284-289)

3)    What are the criteria to diagnose active and inactive sacroiliitis?

- Thanks for your comment.

- According to the 2009 ASAS criteria, active sacroiliitis in SpA is defined by the presence of either one bone marrow edema lesion.

- “ In this criteria, active sacroiliitis in SpA is defined by the presence of either one bone marrow edema lesion on two consecutive MRI slices or ≥2 lesions on a single slice (2, 5).”(in line 101-103)

4)    How new ASAS classification was helpful in diagnosis of axSpA disease?

- Detection of radiographic sacroiliitis is complicated. Several studies showed inter-reader variability, high measurement error. Also, previous criteria is not suitable for the early diagnosis and management of SpA. 2009 ASAS criteria comprises of genetic and imaging arm, which led to classify patients lacking radiographic changes on X-ray (nr-axSpA).

- “When structural abnormalities become apparent on X-ray, patients with AS typically had already symptoms, such as pain, for several years (5). Also, detecting radiographic sacroiliitis is complicated because of inter-reader variability, high measurement error, and low signal-noise ratio (10). It is well known that detecting radiographic sacroiliitis (sacroiliac joint inflammation) in conventional X-ray is crucial but complicated, leading to the delay of diagnosis in AS. But as magnetic resonance imaging (MRI) has been used as an additional imaging arm for diagnosing axSpA by 2009 ASAS criteria, sacroiliac joint inflammation can be used as an early diagnostic marker (2).”(in line 88-95)

5)    Reference must be provided for Symbia Intevo16 hybrid SPECT/CT system

- Thanks for your comment. I cited reference at line 155 (Ref. 17, 18).

 “ Towards standardization of absolute SPECT/CT quantification: a multi-center and multi-vendor phantom study. Peters SMB, van der Werf NR, Segbers M, van Velden FHP, Wierts R, Blokland KJAK, Konijnenberg MW, Lazarenko SV, Visser EP, Gotthardt M.EJNMMI Phys. 2019 Dec 26;6(1):29..”

6)    What is the role of MDP. Why and how much concentration was injected to the axSpA. Incorporate information with references

- Thanks for your comment. I cited references at line 157-160.

- The phosphate groups of Tc-99m MDP binds strongly to metabolically active sites in bone. This yields scitigraphic uptake studies. The role of MDP is described in methods and the role in axSpA is described in introduction.

- “The phosphonate groups on Tc-99m MDP bind firmly to hydroxyapatite crystals in bone through chemisorption (19). Because of its high affinity to metabolically active sites in bone, Tc-99m MDP is commonly used in scintigraphic uptake studies to identify the lesion (20).”

- “Axial inflammation, bone destruction, and new bone formation are critical events in the pathophysiology of axSpA (5). A tracer such as Tc-99m, which shows a high affinity to metabolically active sites, can provide additional information in axSpA in line 106-108”

7)   It is mentioned that the additional information other than SIS ratios from SPECT/CT were not investigated in the said study. What are those factors and why those factors were not considered. Please justify.

- Thanks for the important point.

- No additional information such as osteosclerosis other than the SIS ratio obtained from SPECT/CT was investigated in this study. This is to compare only the factors that rheumatologists can immediately use in clinical practice, but these limitations were re-written in columns 305-315 as limitations of this study in the discussion part.

8)    What was the statistics used to decide the number of control and diseased subjects. Which ratio gives the best results. In the said study diseased samples were 40 whereas control samples were 403 (table1). Please justify the number of samples (disease and control) considered along with references.

- Thanks for the important point.

- This study was not a prospective study but a retrospective chart record analysis study, so it was impossible to obtain the number of samples with statistical significance. The significantly small number of analyses in the SpA group is a limitation of this study (line 298-303).

The final number of SpA patients for analysis yield needs to be increased to make a firm conclusion.

Results:

  • Sacroiliac joint and sacrum must be labelled in Fig1 in order to have clear understanding of the score (0,1,or 2). How sacroiliitis was identified in SPECT/CT appropriately.

- Thanks for the important point.

- Score of 0,1,2 is labelled each in figure 1.

2)    It is written “A total of 443 patients were included”. However table 1 shows AxSpA (n=40) indicating diseased case is 40 and Control subjects were n=403.  What were the criteria considered for control sample. Further, total number of cases in SPECT/CT, MRI, and sacroiliac X-ray images is different. Author must clarify about the total number of cases and control.

 - Thanks for the important point.

- Among the 443 cases which took SPECT/CT scans with LBP, axSpA was diagnosed in only 40 patients. (in line 195-196)

3)    The data (table1) shows all male subjects. Why females were not included in the study. Female participants must have been involved. Please include RF, duration of disease, medication etc in the table.

   - I am so sorry for making confusion. Table-1 includes both sex. It just presents male’s ratio.

   - Table 1 was reorganized with non-parametric tests and descriptive statistics, and the changed parts were re-described in the text and table1. We added male and female ratios respectively. We apologize for not adding rheumatoid factor or current drug use in the previous data review because they were not investigated.

4)    The disease duration is not mentioned. Therefore the data ie “duration of pain in months” is not clear. Pain must have been mentioned in number of days per year. Please clarify.

  - Thanks for your comment. I clarified that present the duration of pain was months in line 131.

5)    Comparison of data between SPECT/CT and MRI must be have been shown in table format for more clear analysis.

- Thanks for the important point.

- Table 3 is a table showing valid statistical values for diagnosis by correcting the SIS ratio of MRI and SPECT/CT within valid performance values. This is a statistical analysis with missing values, so it is described as a table after verification by a statistician.

6)    Figure showing MRI evidenced bone marrow edema could have been incorporated.

- Thanks for the important point.

-As described in the method, the MRI interpretation was not performed by a rheumatologist, but a review of the MRI image reading by a radiologist. Therefore, there is no image fusion of MRI bone marrow edema in the SPECT/CT of Figure 1.

Discussion:

  • In axSpA, apart from Sacroiliac joints there are other responsible factors causing axSpA and SPECT/CT bone imaging identifies sacroiliitis in axSpA patients. Sufficient discussion is needed in this regard along with references.

- Thank you for this comment.

- We add the sentence “ As SPECT/CT serves as a complementary imaging modality for patients with various diseases, there is an increasing need for quantitative or semi-quantitative SPECT/CT. When this was used to explore osseous metabolism, data derived via simple visual and quantitative analyses (SUVs for the latter) differed (27).” In line 257-260.

2)    Utility of sacroiliac joint/sacrum (SIS) ratios for the disease management of axSpA must be discussed elaborately along with references.

 - Thanks for your comment.

 - We emphasize the sentence “ Kim et al. used SPECT/CT to derive SIS ratios, and successfully detected subtle changes in early axSpA with sensitivity of 80.0%, and specificity of 84.6% (in contrast to bone scintigraphy) (21). Another study found semiquantitative analysis of SPECT/CT using SIS ratio can be an useful predictor in identifying early axSpA patients who are at high risk of structural progression (28). In assessing activity of sacroiliitis with SPECT/CT, Lee et al found SIS ratio showed more favorable results than SUV parameters such as SUVmax (29).” In line 265-271.

3)    What are SUV parameters and how does SUV parameters showed more favourable results than SIS ratio. Please discuss.

- Thanks for the important point.

  - We regret that it is difficult to discuss because SUVs are not included in our study. Therefore, this was described in the limitations part of the study.(in line 314-316)

4)   Correlation between the lab clinical data (ESR, CRP), clinical symptoms (numbness, Claudication etc) and SIS ratios of SPECT/CT visual scoring could have been generated for the enhancement of axSpA diagnosis.

  - Thanks for the important point.

  - According to the results of Table 1 and Table 2, there was no difference between the two groups in ESR and CRP, and there was no difference in symptoms of numbness and claudication.

5)    The advantage/disadvantage of SPECT/CT visual scoring method must have been discussed elaborately along with references.

- Thanks for the important point.

  - As described in the results of the abstract on lines 61 to 65, SIS of SPECTCT has peaks that rheumatology clinicians can distinguish relatively easily, but in this study, it did not show as much diagnostic accuracy as MRI. However, it shows the possibility of an alternative tool for identifying axSpA in real practice.

Conclusion:

The study objective was diagnostic performance of SIS ratios of SPECT/CT in axSpA. But, very less number of patients were considered. Comparative analysis was not clear, not validated, and hence conclusion is not sufficiently strong.

  • Thank you for this comment. This limitation is added to the revised version.
  • “Our study was rather modest number of retrospective medical record analysis from single center, so it was impossible to obtain the number of samples with statistical significance.”(in line 299-301)

Reviewer 2 Report

1.      2. Methods. 2.1. Patient recruitment: “A total of 443 patients were included.” – this information is a result, not a method. It is already mentioned in the Results section, so please delete it from the Methods section.

2.      2. Methods. 2.1. Patient recruitment: “patients were not only from the rheumatology department, but from all departments” - what kind of patients were they? In-patients (admitted to the hospital) or out-patients (examined in the out-clinic)? This information should be present in the text, since patient profile is important for the severity of symptoms.

3.      2. Methods. 2.1. Patient recruitment: “C-reactive protein (CRP) level, and HLA B27 status” – where were these tests done? What is the time relation to the SPECT/CTs (were they done on the same day as the SPECT/CT)? Where these tests done by the same laboratory for all patients? These details are highly significant and should be present in the text, since their variability would potentially induce variability of SpA classification.

4.      2. Methods. 2.1. Patient recruitment: “The Bath Ankylosing Spondylitis Disease Activity Index (BASDAI) score was analyzed in axSpA” – please add the BASDAI’s reference article.

5.      2. Methods. 2.2. Clinical Assessment: “The diagnosis of active sacroiliitis on MRI was adopted based on the new ASAS classification criteria.” – please add a reference for these criteria.

6.      2. Methods. 2.2. Clinical Assessment: “MRI and conventional sacroiliac X-ray were evaluated by the medical re-view, which radiologists confirmed.” – there are no details in the Methods text regarding when, how, where, with what did patients underwent MRI and X-rays.

7.      2. Methods. 2.2. Clinical Assessment: “MRI and conventional sacroiliac X-ray were evaluated by the medical re-view, which radiologists confirmed.” – what does “evaluated by the medical review” mean? Did you mean the medical reviewer? The rheumatologist? It is not clear from the text.  You might want to rephrase.

8.      2. Methods. 2.4. Statistical analyses: “Data are presented in Tables and Figures.” Of course! It is self-evident. There is no need to mention this.

9.      2. Methods. 2.4. Statistical analyses: “Continuous parameters were compared using the Mann-Whitney U-test” and “data are expressed as medians (Q1, Q3) for continuous variables” – using a Mann-Whitney U test and expressing continuous variables as medians indicates that these variables were non-normally distributed. Where they? How was normality of distribution evaluated? Please update the text.

10.  2. Methods. 2.4. Statistical analyses: “data are expressed as medians (Q1, Q3) for continuous variables” – first, maybe the readers will not know what Q1 and Q3 mean, so you should write “quartile”; second, you may want to report IQR (interquartile range) which is the difference of Q3 and Q1; third, this information is not consistent with the Results text and the tables, were continuous variables, such as age and pain duration, are expressed as medians with standard deviations. So, Methods do not agree with Results. Please update the text (Methods, Table content and Table footnotes). See also comment on Table 1.

11.  Table 1 and Table 2:

a.       Update the way in which your report your continuous variables according to the Methods section and Table footnotes (please see previous comment);

b.      “Sex” and “ever smoker” were probably evaluated between subgroups with chi-square test – if so, please add this information in the Table footnotes for consistency;

c.       ESR: please add measurement scale (mm/h);

d.      ESR: please add the way it was reported, mean SD or median IQR;

e.       ESR: it seems it is reported as “mean SD”, but the SD (33.45) is higher than the mean (32.54) in the AxSpA subgroup, which means ESR is not normally distributed, so please report it as median with IQR and update Methods and Table footnoted accordingly if necessary.

f.        CRP: please add measurement scale (mg/L for example);

g.      CRP: please add the way it was reported, mean SD or median IQR;

h.      CRP: please report “IQR” or “minimum-maxim” instead of “Q1, Q3” and update Methods and Table footnoted accordingly.

i.        Footnote: “p <0.05*; p < 0.005**” – since you report the actual p value for each test in the table, mentioning this is completely useless. For example, of course “<0.001**” is “< 0.005**”.

12.  3. Results. 3.1. Patient Characteristics: “72 patients in control group achieved the HLA-B27 test” – the verb “to achieve” is not adequate.

13.  3. Results. 3.1. Patient Characteristics: “AxSpA patients exhibited significant clinical symptoms: peripheral arthritis (p = 0.001), Achilles tendinitis (p < 0.001), enthesitis (p < 0.001), uveitis (p < 0.001), and positive HLA B27 status (p < 0.001) compared to controls.” – AxSpA patients will always have disease manifestations more frequently than controls. In other words, sick people will be more frequently sick than healthy people. Testing and reporting this is useless.

14.  5. Conclusion: “that MRI and SIS ratios of SPECT/CT scanning together may also provide support for the diagnosis of nr-axSpA in the early disease phase” – what do you mean by “that”? The phrase is not clear because of this erroneous connector. Also, why would anybody take a MRI and SPECT/CT “together”? MRI is enough to affirm the diagnosis in clinical context.

Author Response

First of all, thank you for the reviewers’ thoughtful comments. This is the point-by-point answer for the words.

Reviewer 2

  1. Methods. 2.1. Patient recruitment: “A total of 443 patients were included.” – this information is a result, not a method. It is already mentioned in the Results section, so please delete it from the Methods section.

- Thank you for the comment, this sentence is deleted in revised version.(in line 128-129)

  1. 2. Methods. 2.1. Patient recruitment: “patients were not only from the rheumatology department, but from all departments” - what kind of patients were they? In-patients (admitted to the hospital) or out-patients (examined in the out-clinic)? This information should be present in the text, since patient profile is important for the severity of symptoms.

- All patients who visited out-clinic with LBP were evaluated.

- “We initially investigated all out-clinic patients who underwent SPECT/CT for the presence of LBP from August 2016 to April 2020 at a single institution.”(line 127-128)

  1. 2. Methods. 2.1. Patient recruitment: “C-reactive protein (CRP) level, and HLA B27 status” – where were these tests done? What is the time relation to the SPECT/CTs (were they done on the same day as the SPECT/CT)? Where these tests done by the same laboratory for all patients? These details are highly significant and should be present in the text, since their variability would potentially induce variability of SpA classification.

- Thanks for the important point.

- CRP, HLAB27 was done in a single institution. Most of patients conducted HLA B27 and CRP within the 14 days of SPECT/CT.

  1. 2. Methods. 2.1. Patient recruitment: “The Bath Ankylosing Spondylitis Disease Activity Index (BASDAI) score was analyzed in axSpA. ” – please add the BASDAI’s reference article.

- Thank you for the comment.

- Bath Ankylosing Spondylitis Disease Activity Index (BASDAI) score was analyzed in only axSpA. Activity in SpA refers to the inflammation caused by the disease, commonly assessed in daily practice with the BASDAI (1, 5).(line135-137)

  1. 2. Methods. 2.2. Clinical Assessment: “The diagnosis of active sacroiliitis on MRI was adopted based on the new ASAS classification criteria.” – please add a reference for these criteria.

- Thank you for the comment.

- Reference article for the 2009 ASAS criteria is added to the revised version.” The diagnosis of active sacroiliitis on MRI was adopted based on the new ASAS classification criteria (11).” (line 149-150)

  1. 2. Methods. 2.2. Clinical Assessment: “MRI and conventional sacroiliac X-ray were evaluated by the medical re-view, which radiologists confirmed.” – there are no details in the Methods text regarding when, how, where, with what did patients underwent MRI and X-rays.

- Thanks for the important point.

- The final reading of hip joint MRI and conventional sacroiliac X-ray confirmed by radiologists was reviewed by a single rheumatologists to diagnose ax-SpA.

  1. 2. Methods. 2.2. Clinical Assessment: “MRI and conventional sacroiliac X-ray were evaluated by the medical re-view, which radiologists confirmed.” – what does “evaluated by the medical review” mean? Did you mean the medical reviewer? The rheumatologist? It is not clear from the text.  You might want to rephrase.

- Thanks for the important point.

- Final reading of MRI and conventional sacroiliac X-ray confirmed by radiologists were reviewed by a single rheumatologists for the diagnosis of AxSpA.

  1. 2. Methods. 2.4. Statistical analyses: “Data are presented in Tables and Figures.” Of course! It is self-evident. There is no need to mention this.

- Thank you for the comment.

- This sentence is deleted in the revised version. (in line 178-179)

  1. 2. Methods. 2.4. Statistical analyses: “Continuous parameters were compared using the Mann-Whitney U-test” and “data are expressed as medians (Q1, Q3) for continuous variables” – using a Mann-Whitney U test and expressing continuous variables as medians indicates that these variables were non-normally distributed. Where they? How was normality of distribution evaluated? Please update the text.

- Thanks for the important point.

- I am so sorry for making confusion. Table 1 was reorganized with non-parametric tests and descriptive statistics, and the changed parts were re-described and updated in the text and table1.

  1. 2. Methods. 2.4. Statistical analyses: “data are expressed as medians (Q1, Q3) for continuous variables” – first, maybe the readers will not know what Q1 and Q3 mean, so you should write “quartile”; second, you may want to report IQR (interquartile range) which is the difference of Q3 and Q1; third, this information is not consistent with the Results text and the tables, were continuous variables, such as age and pain duration, are expressed as medians with standard deviations. So, Methods do not agree with Results. Please update the text (Methods, Table content and Table footnotes). See also comment on Table 1.

- Thanks for the important point.

- I am so sorry for making confusion. Table 1 was reorganized with non-parametric tests and descriptive statistics, and the changed parts were re-described and updated in the text, table1, and table footnotes.

  1. Table 1 and Table 2:

- Thanks for the important point.

- I am so sorry for bothering and thanks again. Table 1 and 2 are changed parts and re-described following your points.

  1. Update the way in which your report your continuous variables according to the Methods section and Table footnotes (please see previous comment);
  2. “Sex” and “ever smoker” were probably evaluated between subgroups with chi-square test – if so, please add this information in the Table footnotes for consistency;
  3. ESR: please add measurement scale (mm/h);
  4. ESR: please add the way it was reported, mean SD or median IQR;
  5. ESR: it seems it is reported as “mean SD”, but the SD (33.45) is higher than the mean (32.54) in the AxSpA subgroup, which means ESR is not normally distributed, so please report it as median with IQR and update Methods and Table footnoted accordingly if necessary.
  6. CRP: please add measurement scale (mg/L for example);
  7. CRP: please add the way it was reported, mean SD or median IQR;
  8. CRP: please report “IQR” or “minimum-maxim” instead of “Q1, Q3” and update Methods and Table footnoted accordingly.
  9. Footnote: “p <0.05*; p < 0.005**” – since you report the actual p value for each test in the table, mentioning this is completely useless. For example, of course “<0.001**” is “< 0.005**”.
  10. 3. Results. 3.1. Patient Characteristics: “72 patients in control group achieved the HLA-B27 test” – the verb “to achieve” is not adequate.

- Thank you for the comment.

- The verb “achieved” is now changed to “performed” in line 203.

  1. 3. Results. 3.1. Patient Characteristics: “AxSpA patients exhibited significant clinical symptoms: peripheral arthritis (p = 0.001), Achilles tendinitis (p < 0.001), enthesitis (p < 0.001), uveitis (p < 0.001), and positive HLA B27 status (p < 0.001) compared to controls.” – AxSpA patients will always have disease manifestations more frequently than controls. In other words, sick people will be more frequently sick than healthy people. Testing and reporting this is useless.

- Thank you for the comment.

- As data are presented in the table, this explanation is deleted in the revised version.(in line 207-210)

  1. 5. Conclusion: “that MRI and SIS ratios of SPECT/CT scanning together may also provide support for the diagnosis of nr-axSpA in the early disease phase” – what do you mean by “that”? The phrase is not clear because of this erroneous connector. Also, why would anybody take a MRI and SPECT/CT “together”? MRI is enough to affirm the diagnosis in clinical context.

- In this study, SPECT/CT showed high sensitivity and negative predictive value. MRI is still considered as secondary imaging modality for diagnosis of axSpA. But in some patients, with an increased likelihood of axSpA, and when MRI is not available, SPECT/CT can be serve as additional tool. To clarify the meaning, revised version deleted “that MRI”, “Together”.

- Thanks for the important point. We all aree with your opinion.

- For those patients with an increased likelihood, based on clinical considerations such as peripheral arthritis, uveitis, etc., SIS ratios of SPECT/CT scanning may also provide support for the diagnosis of nr-axSpA in the early disease phase.”(line 321-322)

Thank you again for your great efforts.

Round 2

Reviewer 1 Report

1) It is observed that only few incorporation has been highlighted. As the Line number is not mentioned in the revised manuscript hence difficult for smooth re-review

2) The response to the comments No 6 of introduction and No. 2, 6 of Method section, are not sufficiently clear and are not strong.

3) Response to comments 2 of introduction ie” Objective of this study.. not sufficiently clear and strong.  

4) Similarly it is not clear if the matter as per response to 4 of introduction (regarding available marker) is incorporated or not.

5) Insufficient information about “Why and how much concentration of MDP was injected to the axSpA as per comment 6 of method section.

In conclusion author must critically check incorporation of all the response in the manuscript.

Author Response

First of all, I appreciate to give the opportunity and thank you for the reviewers’ thoughtful comments. This is the point-by-point answer for the words.

Reviewer 1

1) It is observed that only few incorporation has been highlighted. As the Line number is not mentioned in the revised manuscript hence difficult for smooth re-review

Answer) We apologize for any inconvenience. At the time of writing the manuscript, it was contributed as a word file with line numbers, but it seems to have been converted into an edited form in the system to the reviewer. This time, I will answer with these points in mind.

2) The response to the comments No 6 of introduction and No. 2, 6 of Method section, are not sufficiently clear and are not strong.

Answer) Thank you for the comment.

-No 6 of introduction:

Quickly checking inflammation with imaging modality is more meaningful for early diagnosis of SpA rather than pain control. This study aims to investigate the utility of sacroiliac joint/sacrum (SIS) ratios of SPECT/CT in diagnosing axSpA in patients with LBP compared to MRI and conventional sacroiliac X-ray examination at a single institution, using a visual scoring assessment by one rheumatologist.

(We mention that in the last paragraph of the introduction. And add as “The main advantages of SPECT/CT are better attenuation correction, increased specificity, accurate localization of disease, and evaluation of the involvement of adjacent tissues.” )

-No. 2, 6 of Method section

: AxSpA is most frequent and important clinical subtype of SpA in real world practice. And since this study is retrospective, it was selected as the clinical group that is easiest to evaluate objectively.

 And nr-axSpA considered an early stage of axSpA. We redefined nr-axSpA in the second paragraph in 2-2 method. (Nr-axSpA is a subtype of axSpA, and it’s classification was based on the MRI or clinical characteristics. )And its clinical meaning described in the discussion with reference (30), (31).

(In the same study, the estimated chance of progression from nr-axSpA to axSpA was over 50 % (30). Nr-axSpA has been considered an early stage of axSpA, but nr-axSpA itself has clinical importance. Disease activity levels and functional impact of the  Nr-axSpA patients are similar to AS patients; the disease burdens look equal (2). It is already known from other studies that nr-axSpA patients showed diagnostic delay compared to axSpA patients (31).)

: The role of MDP is described in 2-3.

The phosphonate groups on Tc-99m MDP bind firmly to hydroxyapatite crystals in bone through chemisorption (19). Because of its high affinity to metabolically active sites in bone, Tc-99m MDP is commonly used in scintigraphic uptake studies to identify the high active lesion (20).

3) Response to comments 2 of introduction ie” Objective of this study…..” not sufficiently clear and strong.  

Answer) Thanks for your comment.

Objective of this study is to investigat the utility of sacroiliac joint/sacrum (SIS) ratios of SPECT/CT in diagnosing axSpA in patients with LBP compared to MRI and conventional sacroiliac X-ray examination at a single institution, using a visual scoring assessment by one rheumatologist.

(in last paragraph in introduction)

4) Similarly it is not clear if the matter as per response to 4 of introduction (regarding available marker) is incorporated or not.

Answer) Thanks for your comment.

In the introductory part, we mentioned the genetic, serological markers and the most important imaging markers, including sacroiliitis.

(Currently, HLA B27 remains the best genetic biomarker for diagnosing AS. C-reactive protein (CRP) measures disease activity, predicting structural progression and therapeutic response (7). Radiographic axSpA (r-axSpA) is usually termed ankylosing spondylitis (AS) (8).)

(It is well known that detecting radiographic sacroiliitis (sacroiliac joint inflammation) in conventional X-ray is crucial but complicated, leading to the delay of diagnosis in AS. But as magnetic resonance imaging (MRI) has been used as an additional imaging arm for diagnosing axSpA by 2009 ASAS criteria, sacroiliac joint inflammation can be used as an early diagnostic marker (2).  In several studies using bone biopsies, animal models, and imaging, the earliest detectable change in biopsy specimens is subchondral bone marrow edema visible on MRI (5).)

5) Insufficient information about “Why and how much concentration of MDP was injected to the axSpA as per comment 6 of method section.

Answer) Thanks for your comment.

In the method 2-3 part, we mentioned the concentrate the MDP.

(Tc-99m methylene diphosphonate (MDP) was injected (740 MBq, 20 mCi) and SPECT was performed 3 h later.)

In conclusion author must critically check incorporation of all the response in the manuscript.

Answer) Thanks for your comment.

In the discussion section, we describe several limitations of this study.( However, the present study only used a tracer uptake of SIS ratio by a visual assessment which could not fully offer the advantage of SPECT/CT.)

Based on this study, I plan a better further study with the help of many reviewer’s advices.

- For those patients with an increased likelihood, based on clinical considerations such as peripheral arthritis, uveitis, etc., SIS ratios of SPECT/CT scanning may also provide support for the diagnosis of nr-axSpA in the early disease phase.”(in conclusion)

Thank you again for your great efforts.
